# Analysis of maternal and child health spillover effects in PEPFAR countries

Gary Gaumer [1] , William H Crown [1] , Jennifer Kates,[2] Yiqun Luan,[1] Dhwani Hariharan,[1] Monica Jordan,[1] Clare L Hurley,[1] Allyala Nandakumar[1]

[1]Institute for Global Health and Development, Brandeis University, Heller School for Social Policy and Management, Waltham, Massachusetts, USA
[2]Global Heallh and HIV Policy, The Kaiser Family Foundation, San Francisco, California, USA

**Correspondence to**
Dr Gary Gaumer;
garygaumer@gmail.com

## ABSTRACT

**Objectives** This study examined whether the US President's Emergency Plan for AIDS Relief (PEPFAR) funding had effects beyond HIV, specifically on several measures of maternal and child health in low-income and middle-income countries (LMICs). The results of previous research on the question of PEPFAR health spillovers have been inconsistent. This study, using a large, multicountry panel data set of 157 LMICs including 90 recipient countries, adds to the literature.

**Design** Seven indicators including child and maternal mortality, several child vaccination rates and anaemia among childbearing-age women are important population health indicators. Panel data and difference-in-differences estimators (DID) were used to estimate the impact of the PEPFAR programme from inception in 2004 to 2018 using a comparison group of 67 LMICs. Several different models of baseline (2004) covariates were used to help balance the comparison and treatment groups. Staggered DID was used to estimate impacts since all countries did not start receiving aid at PEPFAR's inception.

**Setting** All 157 LMICs from 1990 to 2018.

**Participants** 90 LMICs receiving PEPFAR aid and cohorts of those countries, including those required to submit annual country operational plans (COP), other recipient countries (non-COP), and three groupings of countries based on cumulative amount of per capita aid received (high, medium, low).

**Interventions** PEPFAR aid to combat the HIV epidemic.

**Primary outcome measures** Maternal mortality and child mortality rates, vaccination rates to protect children for diphtheria, whooping cough and tetanus, measles, HepB3, and tetanus, and prevalence of anaemia in women of childbearing age.

**Results** Across PEPFAR recipient countries, large, favourable PEPFAR health effects were found for rates of childhood immunisation, child mortality and maternal mortality. These beneficial health effects were large and significant in all segments of PEPFAR recipient countries studied. We also found significant and favourable programme effects on the prevalence of anaemia in women of childbearing age in PEPFAR recipient countries receiving the most intensive financial support from the PEPFAR programme. Other recipient countries did not demonstrate significant effects on anaemia.

**Conclusions** This study demonstrated that important health indicators, beyond HIV, have been consistently and favourably influenced by PEPFAR presence. Child and maternal mortality have been substantially reduced, and childhood immunisation rates increased. We also found no evidence of 'crowding out' or negative spillovers in

## STRENGTHS AND LIMITATIONS OF THIS STUDY

⇒ This study used long baseline and treatment periods (1990–2018) and both a traditional and staggered difference-in-difference (DID) research designs with differing model specifications that included specifications with just baseline covariates to better balance comparison and treatment groups, since randomisation was not feasible.

⇒ The impacts of US President's Emergency Plan for AIDS Relief (PEPFAR) were consistently favourable across six country cohorts we examined separately, though impacts were largest for countries where PEPFAR provided the most aid per capita, and where intense annual planning was required (country operational plans).

⇒ Despite the strength of the study design, there are still potential limitations; though we controlled for numerous baseline country differences, confirmed parallel baselines for treatment and comparison groups, and created estimates using two separate estimation approaches (traditional DID and staggered DID methods), we may have failed to control for other unobserved variables, which may have biased our estimates of the magnitude of PEPFAR impact on various outcomes.

⇒ On the other hand, our comparison group included 18 countries that received very small and infrequent PEPFAR aid over the 2004–2018 period (less than a US$1M in aid over the period, or less than US$0.05 per capita). This may make our estimates of PEPFAR's impact somewhat conservative.

these resource-poor countries. These findings add to the body of evidence that PEPFAR has had favourable health effects beyond HIV. The implications of these findings are that foreign aid for health in one area may have favourable health effects in other areas in recipient countries. More research is needed on the influence of the mechanisms at work that create these spillover health effects of PEPFAR.

## INTRODUCTION AND OBJECTIVES

The US President's Emergency Plan for AIDS Relief (PEPFAR) was authorised in 2003 to address the impact of HIV globally, and has become the largest commitment by any nation to address a single disease (now totaling more than US$100B).[1] PEPFAR was signed into law by President G.W. Bush in 2003 to assist

poor countries in stemming the HIV epidemic, and since then, over 100 countries have received assistance. Unlike many other aid programmes, where funds are transferred to country governments and they execute programmes, the implementation of PEPFAR has always required joint planning and monitoring and partnership with countries.

PEPFAR assists countries by supporting free (to patients) programmes for (1) testing for HIV, (2) treatment for persons living with HIV, (3) providing care and support for families affected by and living with HIV, (4) training new healthcare workers and (5) strengthening domestic health systems in recipient countries, among other services. Programme officials report that PEPFAR has saved 25 million lives, supported 20 million persons with life-saving antiretroviral treatment, and enabled 5.5 million babies to be born HIV-free through the reduction of HIV incidence and prevalence among women and men of childbearing age.[2] Officials say the programme has trained 340 000 health workers, provided critical care and support for 7 million vulnerable children and orphans and their caregivers, and provided services to more than 2.9M girls and young women.[3] Researchers have long confirmed the life-saving effects of PEPFAR activities.[4–9]

During the study period, 2004–2018, PEPFAR disbursed approximately US$70 billion in bilateral aid to more than 100 low-income and middle-income countries (LMICs).[10] While PEPFAR is credited with saving millions of lives and changing the trajectory of the HIV epidemic, there have been concerns and conflicting evidence about other health spillover effects (impacts on outcomes for health conditions other than HIV) of this very large, long lasting and multifaceted programme.

On one hand, there have been concerns about negative spillover effects on other health sectors, given the programme's significant size.[11–13] Some have argued that PEPFAR financing, including for the creation of new jobs, could, theoretically, 'crowd out' the supply of staff and other resources to non-HIV health programmes in poor countries.[14–22] If so, these negative spillover effects could potentially lead to worse, non-HIV related outcomes.[18–20] One such study,[18] using OECD Credit Reporting System data,[23] found financial displacement effects within countries, showing that more HIV/AIDS funding was associated with a reduction in malaria programme funding. Another study found HIV funding was associated with reduced provision of childhood immunizations in sub-Saharan Africa (SSA).[15]

On the other hand, others have posited that PEPFAR could indirectly result in positive health spillover effects in other areas of health, particularly maternal and child health.[17] One confirming study of favourable impacts on maternal care in 257 facilities in eight African countries found evidence that pregnant women who were not living with HIV were more likely to deliver in hospitals in areas that had more HIV-funded patient services or more HIV infrastructure support.[16] Another study examined 46 countries in SSA from 1995 to 2010, and found

3 of 4 vaccination indicators were positively associated with PEPFAR presence.[24] That study found, however, that PEPFAR was not significantly associated with improvements in infant mortality and under-5 child mortality.[15] Another study of under-5 mortality also failed to show significant reductions associated with PEPFAR.[25] One of these studies shows favourable impacts on child mortality by the President's Malaria Initiative in 19 countries in SSA from 2006 to 2014, but the PEPFAR covariate was found to be insignificant in influencing child mortality.[26] However, in a recent Kenya study of prevention of mother-to-child transmission (PMTCT) impacts, the authors found favourable infant mortality declines associated with PEPFAR.[27]

Finally, two early African studies on spillover health indicators show inconclusive results. In a 2000–2005 study of all African countries, 13 health indicators (vaccination rates, life expectancy at birth, sanitation, among others) showed no differences in trends between PEPFAR focus countries, and others.[28] A second early (2003–2008) Demographic and Health Surveys study of adults in nine PEPFAR focus countries (compared with 18 other countries in Africa) showed that all-cause mortality reductions attributed to PEPFAR were larger than the number of attributed HIV deaths. This difference, which would suggest spillovers in lives saved due to PEPFAR, were not statistically significant and deemed inconclusive by authors.[6]

These inconsistent findings of spillover effects on health warrant further analysis. Spillover significance, and direction are critical to health aid policy-making in the future. Following much of the extensive literature about spillovers of PEPFAR, we used several maternal child health (MCH) indicators to examine three research questions, using difference-in-differences (DID) analysis: (1) Do patterns of favourable MCH spillovers exist in a large group of 90 recipient countries over the PEPFAR operating period of 2004–2018? (2) Is the pattern of spillover effects across subgroups of countries consistent with differing intensities of PEPFAR planning and funding across recipient countries? (3) Are there any indications of negative spillover effects of PEPFAR?

## Background on possible spillover mechanisms

While PEPFAR was not intended to directly fund specific non-HIV maternal and child health services, its potential impact in this area seems likely, as reflected in some prior studies.[17 24 29 30]

This impact is possible given its significant investments, totaling more than US$1 billion per year, in the health workforce, laboratory services and various aspects of health systems strengthening.[11 31 32] These investments, in turn, may have had favourable health impacts through healthcare worker training (340 000 health workers trained), improved management, infrastructure, equipment and supply chains, and may have even improved the operational culture and service quality in health facilities.[8] Further, PEPFAR supported creation of new medical education systems, a departure from its core areas of

direct intervention in service delivery, and this probably 'yielded large positive spillover benefits beyond HIV'.[4] PEPFAR has also increasingly emphasised serving women at point of entry for prenatal care, as well as where they seek immunisations for their children, allowing possible spillovers in this area.[11 12 33]

In addition, PEPFAR has supported the provision of HIV treatment, prevention and testing services by supporting clinics and testing sites in previously medically underserved areas, and by so doing, exposed many households to professional medical care services for the first time. Indeed, emerging evidence from activity-based costing and management studies across multiple countries indicates that PEPFAR-funded clinicians, who are fully funded to do HIV work, are able to spend 30%–40% of their time providing broader primary healthcare services at the facility level.[34 35] It seems likely that many or most of these non-HIV services would not have been provided to households in the absence of PEPFAR.

Finally, PEPFAR spending has also boosted the economy and jobs in recipient countries, as found by prior research.[36 37] This economic spillover may have created better healthcare access for households and contributed to other factors that could improve general health.

## METHODS

### Study design

Using a 29-year panel dataset (1990–2018) of 157 LMICs, we assessed the spillover impacts of PEPFAR on maternal and child health indicators. Ninety of these countries were PEPFAR recipients. The analysis makes estimates of PEPFAR impacts for six recipient cohorts of countries, each representing different intensities of PEPFAR funding and programme oversight. These cohorts are (1) all 90 recipient countries taken together, (2) the 31 countries where countries participate in an annual country operational planning (COP) process with PEPFAR/OGAC staff, countries which generally have received more aid and, in many cases, have greater HIV burden, (3) other 59 recipient countries that do not prepare COPs, (4) the 30 countries with the highest cumulative PEPFAR aid (per capita) over the 2004–2018 period; (5) the 30 countries with the lowest cumulative aid, and (6) the 30 countries with medium levels of cumulative aid. As we might expect with all aid programmes, the early recipients were generally those with the highest HIV prevalence and the use of these cohorts, each studied separately, avoids the issue of endogeneity that might occur in pooling countries.

We assessed PEPFAR's impact by comparing country cohort outcomes to a comparison group of 67 LMICs that either did not receive any PEPFAR support or received minimal PEPFAR support (<US\$1M in total or <US\$0.05 per capita cumulatively between 2004 and 2018). Online supplemental file S1 provides lists of countries for each cohort and the comparison group.

**Table 1** Indicators analysed in this study

| Variable | Definition |
|---|---|
| DPT immunisations | Per cent of children ages 12–23 months who received DPT vaccinations (3 doses) |
| HepB3 immunisations | Per cent of children ages 12–23 months who received hepatitis B vaccines (3 doses) |
| Measles immunisation | Per cent of children ages 12–23 months who received the measles vaccination |
| Newborns protected against tetanus | Percentage of births by women of childbearing age who are immunised against tetanus |
| Maternal mortality ratio | Number of women who die from pregnancy-related causes while pregnant or within 42 days of pregnancy termination per 100 000 live births |
| Child mortality rate | Probability of a child dying between birth and 5 years of age, per 1000 live births |
| Prevalence of anaemia among women of reproductive age | Prevalence of anaemia among women of reproductive age (% of women ages 15–49) |

DPT, diphtheria, pertussis and tetanus; HepB3, immunisation against acute hepatitis B.

### Outcome variables

Seven indicators of maternal and child health were selected from the World Bank's World Development Indicators (WDI)[38] list. These indicators include four child immunisation rate indicators (diphtheria, pertussis and tetanus (DPT), hepatitis B, and measles among children under age 5; newborns protected against tetanus), the maternal mortality rate, child mortality rate and the prevalence of anaemia among women of reproductive age (which is a risk factor for poor maternal and child birth outcomes). These indicators were selected because the child health indicators have been used in the literature to study PEPFAR spillover impacts and the maternal indicators are commonly used as indicators of maternal health. These indicators are defined in table 1.

### Covariates

We accounted for country-level time-invariant covariates to balance the characteristics between PEPFAR and comparison countries. Our emphasis on baseline covariates rather than time-varying covariates was a deliberate decision that was made to avoid endogeneity problems. Causal inference regarding PEPFAR's impact on spillovers is derived from a model specification designed to implement the Rubin potential outcomes framework.[39]

We intentionally used countries' baseline covariates in 2004 to avoid endogeneity issues. The covariates include gross domestic product per capita, total population, life expectancy at birth, fertility rate, per cent urban population of total population, gross enrolment rate for secondary education, HIV prevalence among the population ages 15–49, non-PEPFAR donor spending on health per capita, domestic health spending per capita, a dichotomous variable indicating whether a country received HIV funding from the USA prior to 2004, and a dichotomous variable indicating country income classifications. We also controlled for the prevalence of diphtheria, pertussis, tetanus (DPT), hepatitis B and measles among the under-5 population when examining DPT, hepatitis B and measles immunisation rates as outcomes in our regressions. We obtained data on included covariates from five publicly accessible databases: the World Bank's WDI,[38] U.S. government's foreignassistance.gov database,[40] OECD Creditor Reporting System,[23] the United Nations Population Division[41] and the Institute of Health Metrics and Evaluation GBD Result's Tool.[42] The rationale for including each covariate is specified in online supplemental file S2.

## Statistical analysis

We employed both the traditional DID method (traditional DID) and a staggered DID panel event study approach (staggered DID) to estimate PEPFAR's impact on our selected MCH indicators. The traditional DID approach assesses the PEPFAR impact in recipient countries after PEPFAR began operating in 2004 and thereafter. The staggered DID approach refines the DID estimates to account for the country-specific year in which PEPFAR began to fund country operations.

For both traditional and staggered DID, in addition to using equations (1) and (3) below with adjusted covariates, we also estimated PEPFAR's impacts without adjusting for baseline covariates. Robust SEs were calculated for all estimations. The full estimation results are shown in online supplemental files S2–S6.

### Traditional DID

The traditional DID method has been widely used in the programme evaluation literature to estimate treatment effects as a non-parametric alternative to parametric sample selection models.[43] The method can be used to estimate attribution to a treatment intervention, such as PEPFAR funding, using preintervention and postintervention data for both treatment and non-treatment countries (eg, the comparison group). The traditional DID method creates an impact estimate by computing the pre–post change in the treatment group and subtracting from it the pre–post change in the comparison group. Econometrically, we used equation (1) to estimate PEPFAR's impact with the traditional DID method:

$$MCH_{it} = \beta_0 + \beta_1 PEPFAR_i + \beta_2 POST_t + \beta_3 PEPFAR_i \times POST_t + \beta_4 X_{i,2004} + \varepsilon_{it} \quad (1)$$

where $MCH_{it}$ is the outcome indicator for country $i$ in year $t$; $PEPFAR_i$ is a dichotomous variable that equals one if country $i$ is a PEPFAR recipient country and zero otherwise; $POST_t$ is a dichotomous variable that equals one for the year $\geq 2004$ and zero otherwise; $X_{i,2004}$ is a set of country-level covariates in the baseline year 2004; $\varepsilon_{it}$ is an error term. The key parameter of interest is $\beta_3$, representing the difference in pre–post changes between PEPFAR countries and comparison countries.

The assumption of parallel trends in outcomes before the intervention is essential for the internal validity of the traditional DID method. We assessed this assumption by examining the descriptive levels and trends of each outcome between PEPFAR and comparison countries at the aggregate level and empirically regressing the following equation:

$$MCH_{it} = \beta_0 + \beta_1 PEPFAR_i + \beta_2 YEAR_t + \beta_3 PEPFAR_i \times YEAR_t + \varepsilon_{it} \quad (2)$$

where $YEAR_t$ entered as a set of dichotomous variables with 2004 as the reference year. The main parameter of interest is $\beta_3$, representing the moderating effect of country's PEPFAR status on the difference in outcomes for a given year relative to 2004. A series of statistically insignificant $\beta_3$ s before 2004 suggested the satisfaction of the parallel trend assumption between PEPFAR and comparison countries. The tests confirmed parallel baselines for every dependent variable for every country cohort (see online supplemental file S7).

### Staggered DID

We used staggered DID to account for when countries joined the PEPFAR programme. The staggered DID approach generalises the traditional DID method by considering country-specific participation year and allows for dynamic leads and lags to the participation to be estimated.[44] Equation (3) was used for staggered DID estimates:

$$MCH_{it} = \beta_0 + \beta_1 PEPFAR_i + \beta_2 D_{it} + \beta_3 X_{i,2004} + \gamma_t + \varepsilon_{it} \quad (3)$$

where $D_{it}$ is a dichotomous variable that equals one for country $i$ in each of its participation years and zero otherwise; $\beta_2$ is of our primary interest, measuring the impact of the presence of PEPFAR's support on recipient country MCH indicators; $\gamma_t$ is a vector of year fixed effects dichotomous variables.

Specification testing of the staggered DID models was accomplished by estimating the dynamic impact of PEPFAR on each MCH outcome (see online supplemental file S8 for the results). This investigation serves two purposes: first, to determine whether PEPFAR and comparison countries had similar preintervention outcome trends before recipient countries joined the programme; and second, to track the evolution of PEPFAR's impact in terms of magnitude and statistical significance over time. We achieve this by substituting $D_{it}$ in equation (3) with a set of country-specific dichotomous variables, $D_{it}^T$, indicating years relative to each country's PEFPAR initial participation year. Equation (4) was used:

$$MCH_{it} = \beta_0 + \beta_1 PEPFAR_i + \sum\nolimits_{T=-10}^{15} \beta_T D_{it}^T + \delta X_{i,2004} + \gamma_t + \varepsilon_{it} \qquad (4)$$

where $D_{it}^T$ equals zero, except as following: $D_{it}^T$ ($-10 \leq T \leq -1$) equals one for country $i$ in its $T$th year before participating the PEPFAR programme, while $D_{it}^T$ ($0 \leq T \leq 15$) equals one for country $i$ in its $T$th year after participating the PEPFAR programme. A set of statistically insignificant $D_{it}^T$ ($-10 \leq T \leq -1$) without a discernible trend would confirm a similar preintervention trend between PEPFAR and comparison countries, demonstrating the internal validity of the staggered DID design. Further, we used the year prior to initial participation as the baseline year and therefore excluded the dummy variable of $D_{it}^{-1}$ from the equation. We followed previous studies[45] in using a 25-year window, which set $D_{it}^{-10}$ equals one for all years that are 10 or more years before PEPFAR participation, while $D_{it}^{15}$ equals one for all years that are 15 or more years after PEPFAR participation. Thus, PEPFAR's dynamic impacts estimated for these two endpoints may have greater variance and, consequently, less precision. Our results showed that in some situations the baseline assumptions of staggered DID were not met. In the case of immunisation outcomes, this assumption was met for HepB vaccination, but only met for DPT, measles and tetanus vaccination in the COP and high spending cohorts. While child mortality and anaemia outcome models met the assumption, maternal mortality generally did not meet the assumption.

Lastly, we conducted two types of robustness tests. First, we did a placebo test to examine the robustness of PEPFAR impacts estimated from the staggered DID design. To conduct the test, we randomly assigned each recipient country a pseudo-policy-start year which is between 1990 and the country's actual PEPFAR participation year and used these randomly assigned years to run equation (3). We repeated the process 500 times. We examined the kernel density distribution of the 500 estimated $\beta_2$ s against a normal distribution and plotted p values of these 500 $\beta_2$ s. A kernel density distribution with a mean close to zero and p values being larger than 0.05 is indicative of good robustness of our staggered DID estimates with country actual participation years. Online supplemental file S9 shows that for MCH outcomes the mean of the estimated $\beta_2$ s with pseudo-policy-start years is close to zero, with the majority of the associated p-values being greater than 0.05. This suggests that the estimated PEPFAR impacts calculated using the actual PEPFAR participation years are not obtained by chance, confirming the robustness of the staggered DID estimates. Second, we logarithmically transformed the six outcomes and non-dichotomous covariates and regressed both the adjusted and unadjusted equations (1) and (3). We compared coefficients in terms of sign and significance with our non-logged estimates which are reported in tables 2 and 3. The patterns of sign and significance of PEPFAR's impact in the logged models closely mirror the unlogged results reported in tables 2 and 3 (online supplemental file S10).

We used Stata V.18.0 (StataCorp LLC) for all analyses.

## RESULTS

### Impact of PEPFAR on the four immunisation rates

Figure 1 presents the levels and trends in the four immunisation rates from 1990 to 2018 across all PEPFAR countries, COP countries and comparison countries. PEPFAR recipient countries, particularly the COP countries, initially had lower (poorer) values for all four immunisation rates compared with the comparison countries, but they demonstrated increasing immunisation rates over time. In all cases, however, the improvement appears to be faster in PEPFAR countries than in comparison countries. The post PEPFAR uptake for DPT and measles is more pronounced than hepatitis B, and the situation is even less pronounced for tetanus than hepatitis B. In all cases, the preperiod trend pattern (before 2004) is generally the same in PEPFAR countries and comparisons.

Pre-2004 parallel trends are observed for all immunisation rates and country cohorts in the traditional DID setting, with most 95% CIs of $\beta_3$ in equation (2) before 2004 being insignificantly different from zero. However, for staggered DID, the satisfaction of the essential pretrend assumption was generally met, but with exceptions. Specifically, all the six cohorts fulfilled the assumption for the HepB rate, while only the COP and high spending cohorts satisfied the assumption for DPT, measles and tetanus (see online supplemental files S7 and S8 for detail).

Table 2 reports the key estimation results from traditional DID ($\beta_3$ in equation 1) and staggered DID ($\beta_2$ in equation 3) for the six country cohorts. For each cohort of countries (six columns), we report four models for each of the four immunisation rate outcomes. The models include no covariates (model 1) and covariates (model 2) versions of the traditional and staggered DID methods.

Specifically, both econometric methods (traditional DID and staggered DID) suggest that PEPFAR's support has been associated with improvements in DPT, measles and tetanus immunisation rates for all the six country cohorts relative to what would have been expected in the absence of the programme. The coefficients are interpreted as the changes in the dependent variable attributed to PEPFAR presence through 2018. Generally, the estimated coefficients are similar in magnitude, positive sign and statistical significance across the adjusted and unadjusted specifications.

The single exception is that in one of the models for tetanus the coefficients for (model 2 (adjusted) using the staggered DID specification), shows a positive coefficient for all cohorts, but the coefficient is insignificant for the cohort receiving the least amount of PEPFAR support.

For the hepatitis B immunisation rate, PEPFAR's support results are somewhat inconsistent, with only 14 of 24 PEPFAR coefficients being positive and significant. But, the adjusted models generally show (10 out of 12 models) strong favourable PEPFAR impacts across

**Table 2** PEPFAR impacts on four immunisation rates: traditional and staggered DID estimates

| Outcomes and estimation methods | All PEPFAR countries | COP countries | Non-COP countries | High-intensity PEPFAR countries | Medium-intensity PEPFAR countries | Low-intensity PEPFAR countries |
|---|---|---|---|---|---|---|
| **DPT immunisation rate** | | | | | | |
| Means | 78.0 | 74.2 | 79.9 | 74.8 | 80.1 | 79.0 |
| Model 1. Unadjusted model (traditional) | 8.421*** | 8.607*** | 8.221*** | 9.728*** | 5.467*** | 9.855*** |
| Model 1. Unadjusted model (staggered) | 8.206*** | 6.538*** | 10.80*** | 8.373*** | 4.149** | 14.49*** |
| Model 2. Adjusted model (traditional) | 8.810*** | 8.693*** | 8.889*** | 9.380*** | 6.635*** | 10.07*** |
| Model 2. Adjusted model (staggered) | 7.328*** | 7.459*** | 8.334*** | 8.326*** | 6.836*** | 8.979*** |
| **Hepatitis B immunisation rate** | | | | | | |
| Means | 79.1 | 74.4 | 81.7 | 80.7 | 78.3 | 78.5 |
| Model 1. Unadjusted model (traditional) | 1.374 | 8.027** | −2.097 | 2.661 | 2.163 | −0.648 |
| Model 1. Unadjusted model (staggered) | 5.436** | 16.33*** | 3.000 | 10.78* | 3.096 | 6.397 |
| Model 2. Adjusted model (traditional) | 6.667** | 13.64*** | 3.638 | 8.922** | 5.446 | 7.410* |
| Model 2. Adjusted model (staggered) | 10.52*** | 21.33*** | 7.050*** | 17.15*** | 6.967** | 9.912*** |
| **Measles immunisation rate** | | | | | | |
| Means | 77.5 | 74.5 | 79.1 | 72.9 | 79.9 | 79.7 |
| Model 1. Unadjusted model (traditional) | 6.644*** | 7.192*** | 6.260*** | 7.034*** | 4.287** | 8.439*** |
| Model 1. Unadjusted model (staggered) | 6.816*** | 5.091*** | 9.077*** | 5.692*** | 3.280* | 13.28*** |
| Model 2. Adjusted model (traditional) | 6.946*** | 7.462*** | 6.560*** | 6.973*** | 5.061*** | 8.529*** |
| Model 2. Adjusted model (staggered) | 5.515*** | 5.546*** | 6.115*** | 5.158*** | 5.357*** | 7.083*** |
| **Prevalence of newborns protected against tetanus** | | | | | | |
| Means | 75.7 | 75.6 | 75.8 | 75.1 | 78.3 | 74.2 |

Continued

**Table 2** Continued

| Outcomes and estimation methods | All PEPFAR countries | COP countries | Non-COP countries | High-intensity PEPFAR countries | Medium-intensity PEPFAR countries | Low-intensity PEPFAR countries |
|---|---|---|---|---|---|---|
| Model 1. Unadjusted model (traditional) | 6.069*** | 4.775** | 6.995*** | 6.154*** | 5.845** | 6.059*** |
| Model 1. Unadjusted model (staggered) | 5.218*** | 5.444*** | 5.942*** | 7.005*** | 4.966** | 5.401*** |
| Model 2. Adjusted model (traditional) | 5.240*** | 3.736* | 6.393*** | 5.271*** | 5.510** | 5.229*** |
| Model 2. Adjusted model (staggered) | 3.619*** | 4.053** | 4.103*** | 5.475*** | 5.391** | 1.022 |

*p< 0.05; **p < 0.01; ***p < 0.001.
COP, country operational plans; DID, difference-in-differences; DPT, diphtheria, pertussis, tetanus; PEPFAR, President's Emergency Plan for AIDS Relief.

cohorts, with larger coefficients for the staggered DID compared with the traditional DID. This pattern of somewhat inconclusive findings is suggested in the trend data in figure 1. All the coefficients for the adjusted staggered model, and four of the six coefficients for adjusted model for traditional DID are significant (the exceptions are the cohorts of the non-COP countries and the cohort of medium levels of PEPFAR funding).

Other consistent patterns are evident in table 2. Excepting the hepatitis B immunisation rate, model 1 (no baseline covariates) has coefficients that are similar in size when compared with model 2 (with covariates). And results show consistency in size and significance of coefficients across cohorts, except for funding intensity: with medium-intensity cohorts, PEPFAR generally shows smaller impacts.

### Impacts of PEPFAR on maternal and child mortality rates and anaemia in women

Figure 2 describes the unadjusted trends in these outcomes. The maternal mortality rate for comparison countries remains relatively flat without improvement, whereas rates in PEPFAR countries improve and catch up to the comparison group (figure 2A). The under-5 child mortality rate (figure 2B) shows a very clear pre-2004 pattern of higher child mortality in PEPFAR recipient countries and then faster declines after 2004. In all PEPFAR and COP recipient countries, there were higher pre-2004 levels of under-5 mortality and even faster declines after 2004. Trends show poorer pre-2004 levels of anaemia among women aged 15–49 in all PEPFAR countries (figure 2C), and particularly in the COP country segments, with only slight evidence of convergence between PEPFAR and comparison countries.

All three outcomes in all six cohorts satisfied the parallel trend assumption in the traditional DID setting. However, in staggered DID, the pretrend assumption was not met in certain cases. Specifically, the assumption was met for child mortality rates in COP and high-intensity countries (but not for the other cohorts). The assumption was met in all cohorts for anaemia. Maternal mortality rates did not satisfy the assumption in all country cohorts, indicating the limited internal validity for the maternal mortality results in the staggered DID models. (See online supplemental files S7 and S8 for detail.)

Table 3 shows that maternal mortality ratios and child mortality rates are substantially and consistently lower with the presence of PEPFAR. With the adjusted specifications (model 2), the pattern generally shows larger reductions in maternal mortality ratios and child mortality rates for COP and high-intensity segments. Notably, these two segments also have much higher baseline maternal mortality ratios and child mortality rates compared with other recipient segments. However, table 3 shows that PEPFAR has inconsistent effects on reducing the prevalence of anaemia in women of reproductive age across different country cohorts. A statistically significant spillover benefit is only consistently observed in high-intensity country cohorts across all econometric methods and modelling specifications.

Here too, these modelling results in table 3, excepting the prevalence of anaemia among women, show consistency in PEPFAR influences across models (1 and 2) and across the traditional DID and staggered DID approaches.

Table 3 PEPFAR impacts on maternal mortality ratio, child (under-5) mortality rate, and the prevalence of anaemia among women of reproductive age: traditional and staggered DID estimates

| Outcomes and methods | All PEPFAR countries | COP countries | Non-COP countries | High-intensity PEPFAR countries | Medium-intensity PEPFAR countries | Low-intensity PEPFAR countries |
|---|---|---|---|---|---|---|
| **Maternal mortality ratio** | | | | | | |
| Means | 409.8 | 497.5 | 363.7 | 519.6 | 345.5 | 364.4 |
| Model 1. Unadjusted model (traditional) | −96.46*** | −120.90*** | −83.62* | −125.77*** | −98.65* | −64.96 |
| Model 1. Unadjusted model (staggered) | −109.33*** | −82.58* | −171.61*** | −145.11*** | 19.84 | −310.18*** |
| Model 2. Adjusted model (traditional) | −100.74*** | −130.90*** | −85.40*** | −135.68*** | −100.04*** | −68.94** |
| Model 2. Adjusted model (staggered) | −60.11*** | −80.42*** | −58.73*** | −118.64*** | −53.09* | −95.90*** |
| **Child (under-5) mortality rate** | | | | | | |
| Means | 78.9 | 97.9 | 68.9 | 99.5 | 65.4 | 71.8 |
| Model 1. Unadjusted model (traditional) | −26.90*** | −34.29*** | −23.02*** | −37.76*** | −21.98*** | −20.97*** |
| Model 1. Unadjusted model (staggered) | −22.41*** | −30.81*** | −26.47*** | −36.15*** | −11.73** | −37.12*** |
| Model 2. Adjusted model (traditional) | −27.38*** | −35.67*** | −23.17*** | −40.24*** | −21.81*** | −21.18*** |
| Model 2. Adjusted model (staggered) | −20.65*** | −32.80*** | −18.44*** | −37.75*** | −19.11*** | −19.56*** |
| **Prevalence of anaemia among women of reproductive age** | | | | | | |
| Means | 37.0 | 39.0 | 36.0 | 38.6 | 36.3 | 36.2 |
| Model 1. Unadjusted model (traditional) | −1.26 | −1.91* | −0.91 | −2.39** | −0.90 | −0.48 |
| Model 1. Unadjusted model (staggered) | −1.25* | −1.13 | −1.99* | −1.67* | 1.29 | −4.20*** |
| Model 2. Adjusted model (traditional) | −1.02 | −1.58* | −0.74 | −2.18** | −0.72 | −0.25 |
| Model 2. Adjusted model (staggered) | −0.42 | −1.09 | 0.00 | −1.71* | −0.55 | 0.02 |

*p< 0.05; **p < 0.01; ***p < 0.001.
COP, country operational plans; DID, difference-in-differences; PEPFAR, President's Emergency Plan for AIDS Relief.

## DISCUSSION

Our assessment of PEPFAR's impact on several non-HIV/AIDS health measures in the areas of maternal and child health finds favourable consistent evidence of spillover effects for child immunisation rates, maternal mortality and child mortality. Though immunisations are not provided directly by PEPFAR, utilisation rates for several childhood immunisations demonstrate positive spillover health benefits of PEPFAR presence in recipient countries. We estimate that four childhood immunisations

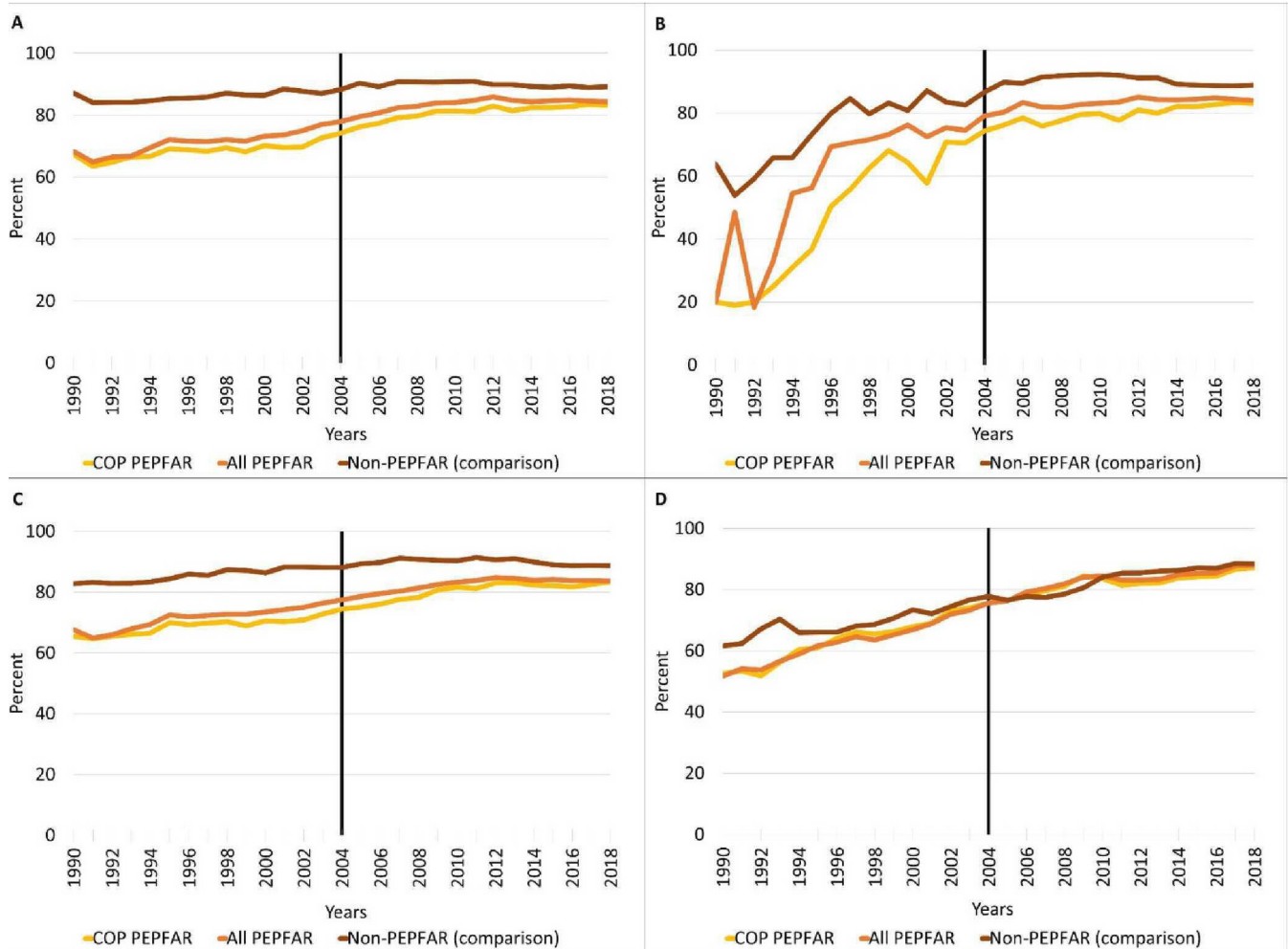

**Figure 1** Trends in childhood immunisations in US President's Emergency Plan for AIDS Relief (PEPFAR) recipient countries: (A) DPT immunisation rate, (B) hepatitis B immunisation rate, (C) measles immunisation rate, and (D) prevalence of newborns protected against tetanus. COP, country operational plans; PEPFAR, President's Emergency Plan for AIDS Relief.

were higher by 2018 than would have been expected in the absence of PEPFAR. The two earlier research studies were conflicting in their findings about immunisation spillovers. Our study confirms one of these earlier and less comprehensive studies showing positive spillovers[24] and conflicts with the other prior study[15] showing negative effects using SSA data from 2003 to 2010.

We find large and significant impacts of PEPFAR on child mortality across all country segments and models. This confirms one earlier study done in Kenya[27] and differs from three earlier studies that all showed insignificant relationships between PEPFAR and child mortality.[24–26] Our findings are particularly strong for child and maternal mortality. Some of these reductions are likely due to reductions in HIV-related mortality, but they appear to extend beyond this. For example, HIV accounted for 5.1% of the estimated 4 million child deaths in SSA in 2003, compared with 1.6% of the estimated 2.8 million child deaths in 2018, a drop that would not fully drive the overall decline in child mortality we estimate.[30]

To our knowledge, PEPFAR effects on maternal mortality rates have not been studied before, though studies have confirmed mechanisms[17] for such effects, and confirmed that PEPFAR has improved obstetric care patterns.[16] Our estimates suggest PEPFAR has reduced maternal mortality. While the staggered DID estimates did not satisfy an important preperiod assumption of the method, the effects of the traditional DID satisfied the parallel baseline assumption, and show large and favourable impacts of PEPFAR. Since HIV is not considered to be a direct cause of maternal mortality, the spillover benefits of PEPFAR appear to be, as for child mortality, very substantial (HIV was estimated to have accounted for 6.4% of maternal deaths in SSA between 2003 and 2009).[30]

We find evidence of favourable impacts on the prevalence of anaemia in women of childbearing age in COP and highly funded countries, but the favourable pattern is not seen in other country cohorts. No negative spillover effects were found for any of the seven outcomes we study here.

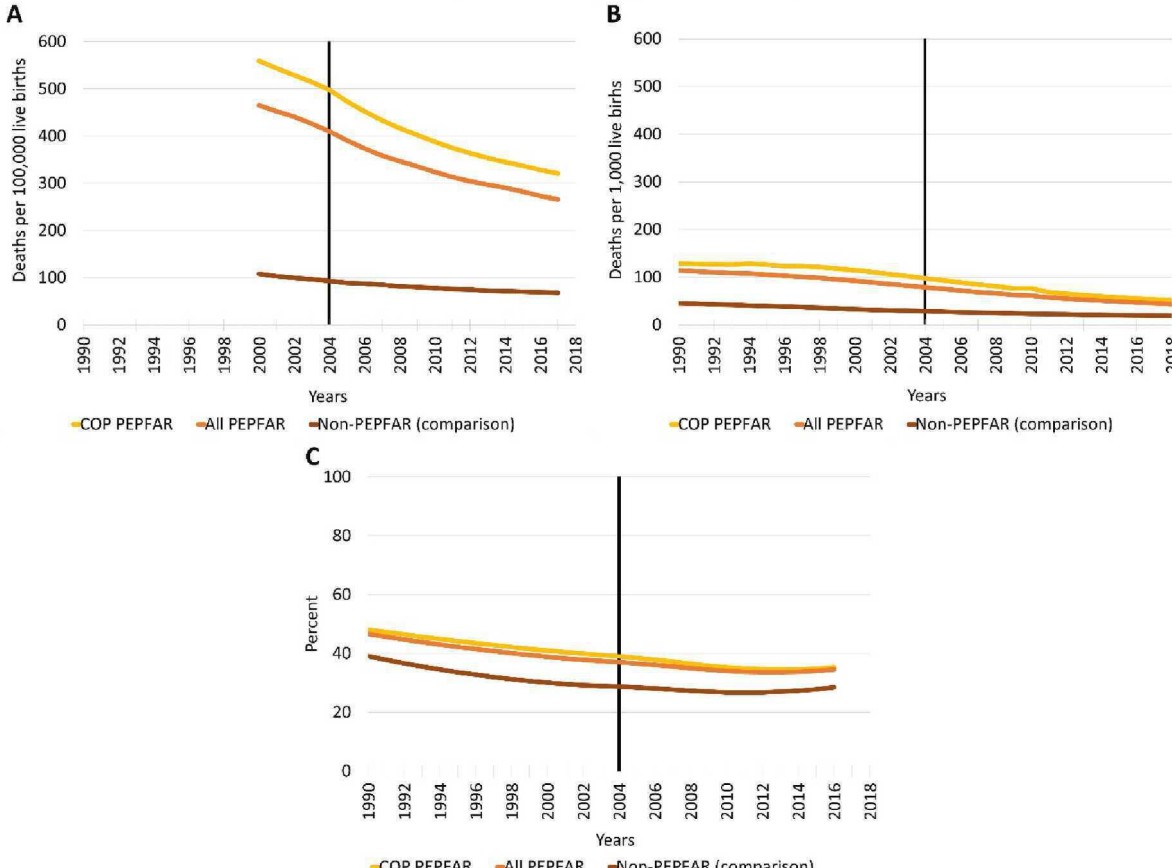

**Figure 2** Trends in PEPFAR recipient countries: (A) maternal mortality ratio, (B) child mortality rate (under-5), and (C) Prevalence of anaemia among women of reproductive age. COP, country operational plans; PEPFAR, President's Emergency Plan fir AIDS Relief.

While our evidence of positive spillover effects on maternal and child health is notable and important, the mechanisms for such effects are not possible to isolate in our research. As noted earlier, one possible means is through PEPFAR's efforts to address HIV in pregnant women and children, since it brings women (and their children) into contact with formal healthcare services and providers.[46] A second, related mechanism may be the widespread scale-up of PEPFAR infrastructure that has created health-delivery system capacities to serve many isolated and previously underserved communities. This scale-up often has been more rapid than the demand for HIV services, enabling and encouraging provision of other needed medical services in these locations.[34] More research is needed to document the source programmatic mechanisms of these and possibly other spillover benefits.

## CONCLUSIONS

In addition to averting HIV deaths and limiting the spread of HIV, we find that PEPFAR funding has had other health benefits beyond HIV for recipient LMIC populations. While some early and focused studies reported instances of PEPFAR 'crowding out' the supply of staff and other resources and evidence of negative spillover effects, this study, which includes a more comprehensive, multicountry data set, with observations obtained over a longer time period, did not identify evidence of any such effects.

This study demonstrates evidence of large, positive spillover effects in maternal mortality and child mortality and also shows consistent evidence of positive spillovers in childhood immunisation rates. Anaemia in women shows favourable impacts but only in the COP and high spend cohorts. These effects may stem from many aspects of the PEPFAR programme, though this study did not examine potential mechanisms for such effects, which warrant further research.

**Acknowledgements** The authors thank Yara Halasa-Rappel, Brandeis University, Waltham, MA, and Adam Wexler, KFF, Washington DC, for data retrieval and dataset preparation.

**Contributors** GG, WHC, JK and AN led the study conception and design. GG, WHC, JK, YL, DH and MJ conducted the analysis and interpretation of results. GG, WHC, JK, YL, DH, MJ and CLH contributed to manuscript preparation. AN was involved in funding acquisition and supervision. All authors reviewed the results and approved the final version of the manuscript. GG is the guarantor of this work.

**Funding** This paper was produced with funding from Palladium International, LLC under subcontract number 217730-Brandeis-01; and Prime Contract number 2021-002516 from the Global Fund to Fight AIDS, Tuberculosis, and Malaria. It was also produced in part with funding from the Bill and Melinda Gates Foundation under grant INV-046299. Its contents are solely the responsibility of Brandeis University and do not necessarily represent the official views of the Bill and Melinda Gates Foundation, Palladium or The Global Fund.

**Competing interests** None declared.

**Patient and public involvement** Patients and/or the public were not involved in the design, or conduct, or reporting, or dissemination plans of this research.

**Patient consent for publication** Not applicable.

**Ethics approval** Not applicable.

**Provenance and peer review** Not commissioned; externally peer reviewed.

**Data availability statement** Data are available in a public, open access repository. Our data came from five publicly available datasets: World Bank's World Development Indicators; U.S. government's foreign assistance.gov database; OECD Creditor Reporting System database; the United Nations Population Division; and the Institute of Health Metrics and Evaluation GBD Result's Tool.

**ORCID iDs**
Gary Gaumer http://orcid.org/0000-0003-3304-1258
William H Crown http://orcid.org/0000-0002-8822-2737

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
