## [Reviewer comments · BMJ Open]

ARTICLE DETAILS

TITLE (PROVISIONAL)	Analysis of Maternal and Child Health Spillover Effects in PEPFAR Countries
AUTHORS	Gaumer, Gary; Crown, William H; Kates, Jennifer; Luan, Yiqun; Hariharan, Dhvani; Jordan, Monica; Hurley, Clare; Nandakumar, Allyala

VERSION 1 – REVIEW

REVIEWER	Richterman, Aaron Hospital of the University of Pennsylvania
REVIEW RETURNED	08-Jun-2023

GENERAL COMMENTS	Authors report a difference-in-difference analysis assessing changes in country-level maternal and child health outcomes associated with a country being a PEPFAR recipient, with primary findings including improvements in vaccine-related outcomes, maternal mortality, and infant mortality. The conceptual justification for measuring spillover effects associated with HIV-related health aid is strong, but I found there to be a number of major issues, primarily methodological but also related to the classification of the study as an economic study and the research gap this study is supposed to be addressing, all outlined in detail below: Title 1. Why is this study called an economic evaluation, and why is the CHEERS checklist used if no costs are assessed? This seems to be an evaluation of health outcomes associated with a health aid program. Introduction 2. Several key references are not included in the introduction or discussion — Bendavid Ann Intern Med 2010 (showing PEPFAR's effects on HIV-related outcomes), and particularly Bendavid JAMA 2012, showing reductions in all-cause mortality with PEPFAR. 3. Apart from those missed references, the authors summarize well the literature assessing the relationship between PEPFAR and maternal / child outcomes. I am left wondering what research gap motivates the current study — is it the size of the study in terms of countries included, or the timing, or something else? I recommend spelling out more clearly what the justification for this study is, particularly given the number of other analyses of country-level outcomes associated with PEPFAR. Methods
--

	4. Related to #3, given prior work evaluating country-level associations between PEPFAR and maternal/child outcomes, this study would be greatly strengthened by considering individual-level outcomes measured by nationally representative surveys such as DHS (data are publicly available upon request). With DHS data, pregnancy-related and infant mortality can be estimated directly (see Bendavid JAMA 2012, Jakubowski PLoS Med 2017, Richterman Nature 2023), and infant vaccination history is also directly assessed. 5. Some countries were selected as PEPFAR focus countries later than 2004, which makes a staggered design preferable, rather than pre-post 2004. 6. There is insufficient detail about the model specification. Are linear models used? Are cluster robust standard errors used? Are country and year fixed effects included? In the supplement, the variables in the equations are not defined. 7. There is no discussion or assessment of the parallel trends assumption, which is key to the difference-in-differences design. 8. Related to #7, temporal analyses (event study estimates) should be included. 9. There have been major advances in the difference-in-differences literature over the last five years, summarized well by Roth et al "What's Trending in Difference-in-Differences? A synthesis of the Recent Econometrics Literature" https://doi.org/10.48550/arXiv2201.01194. In particular there is risk of bias when intervention timing varies (as it does here) and the effect also changes over time. I would recommend including discussion of these issues and consider use of alternate approaches that are not vulnerable to this bias, again well-summarized in the Roth article. 10. Why only consider baseline covariates, rather than time-varying? 11. How was high, medium, and low PEPFAR disbursements defined?
--	--

REVIEWER	Noguchi, Haruko Waseda University, Faculty of Political Science and Economics
REVIEW RETURNED	26-Jul-2023

GENERAL COMMENTS	1. Overall evaluation This study endeavors to assess the spillover effects of the US President's Emergency Plan for AIDS Relief (PEPFAR) on maternal and child health outcomes. To achieve this, the authors employ the difference-in-differences method (DID) on a longitudinal dataset spanning 29 years (1990-2018) comprising 157 low- and middle-income countries (LMICs). Within this dataset, they distinguish 90 PEPFAR recipient countries from 67 countries that either received no PEPFAR support or minimal PEPFAR support. This study provides compelling evidence that the presence of PEPFAR has consistently and positively influenced critical health indicators beyond HIV, including child and maternal mortality rates, as well as childhood immunization rates.
---

Given the inconclusive nature of previous studies regarding the spillover effects of PEPFAR, this study aims to address this knowledge gap by undertaking a comprehensive program evaluation that encompasses seven key health outcome measures. These measures have been thoughtfully selected from the World Bank's World Development Indicators, which serve as a representation of global health target areas. The inclusion of these diverse health indicators enhances the scope and rigor of the analysis, allowing for a more comprehensive assessment of PEPFAR's impact on various health aspects beyond its primary focus on HIV/AIDS.

While I acknowledge the importance of this study, I believe that the current manuscript lacks a comprehensive elucidation of the institutional context of PEPFAR and the underlying hypothetical mechanisms driving the observed spillover effects. Additionally, I would like to highlight certain econometric methodological issues that need to be addressed in this study.

I kindly request you to refer to the detailed comments provided below.

2. Major Comments

2-1. The elucidation of the institutional context and the consequent hypothetical mechanism remains unclear:

Initially, the authors presented a concise overview of PEPFAR and reviewed relevant literature focusing on its spillover effects, summarizing these details in the "Introduction" section. However, for readers who are not well-acquainted with PEPFAR, a comprehensive understanding of the potential mechanisms linking PEPFAR's interventions with other health promotion programs, and thereby leading to unintended and indirect effects (whether positive or negative) on various health outcomes, remains elusive.

For instance, the authors characterized PEPFAR as a "vertical" intervention. This term refers to an approach that addresses a specific health issue in isolation, often through targeted programs. How this would cause "crowd out" of scarce resources in detail? How does PEPFAR contribute to situations where women not infected with HIV end up giving birth in areas with more HIV-funded patient service or greater HIV infrastructure support? Furthermore, there is ambiguity about how PEPFAR facilitates the advancement of vaccination programs or initiatives like the Prevention of Mother-to-Child Transmission (PMTCT) program and the Saving Mothers, Giving Life program.

In light of these concerns, I propose that the authors furnish a more detailed account of PEPFAR, elucidating the hypothetical mechanisms that underpin the observed spillover effects. If the word limit is insufficient, kindly consider relocating all pertinent information of PEPFAR to the appendix section for comprehensive coverage.

2-2. Econometric Strategy: necessary assumptions to be met for DID and timings of intervention:

(1) Initially, when employing a simple DID analysis, adherence to the parallel trend assumption becomes indispensable. Upon meticulous examination of Figures 1 and 2, the presence of this crucial assumption appears inconclusive. To ascertain its validity,

the authors should conduct a rigorous statistical assessment, comparing outcome disparities before the intervention's temporal demarcation in 2004. Specifically, this evaluation should encompass the COP PEPFAR versus non-PEPFAR groups, as well as the ALL PEPFAR versus non-PEPFAR groups.

To accomplish this verification, the authors can undertake the reconstruction of Figures 1 and 2, delineating the distinctions observed in raw data alongside their respective 95% confidence intervals. Crucially, scrutinizing whether these confidence intervals envelop the null point (0) and encompass both positive and negative values prior to 2004 will elucidate the adherence to the parallel trend assumption. Should the confidence intervals encompass 0 and span both directions, it indicates the fulfillment of the parallel trend assumption, denoting the absence of statistically significant outcome disparities between the aforementioned groups preceding the intervention. Conversely, the presence of confidence intervals diverging from 0, while straddling either the positive or negative domains, would signify statistical significance in outcome differences prior to the intervention. Such circumstances would suggest a potential violation of the parallel trend assumption, thereby rendering the results susceptible to bias in the DID analysis.

(2) Secondly, I hold reservations regarding the assumption that the intervention of PEPFAR occurs at random. Despite the authors' efforts to control for various country characteristics during their analysis, I strongly recommend confirming whether the differences in these characteristics between the treatment (PEPFAR-recipient) and control (non-PEPFAR-recipient) groups are statistically insignificant. In essence, it is essential to ascertain if the attributes of both groups are well-balanced. In the absence of balance, adopting a DID combined with a propensity score matching method (PPS) would be prudent. This ensures a more rigorous evaluation of the intervention's impact and mitigates potential biases that might arise from unbalanced characteristics between the two groups.

Therefore, I advise the authors to thoroughly examine and statistically validate the comparability of country attributes between the PEPFAR and non-PEPFAR groups. If imbalances are detected, the implementation of PPS would be a prudent course of action to attain a more robust and credible analysis, thereby enhancing the reliability of the study's outcomes, although PPS could adjust only observable factors.

In addition to the PSM framework, to account for time-invariant unobservable country characteristics, one may consider employing a fixed-effect approach by leveraging the panel data. Furthermore, to address time-variant unobservable country attributes, it is essential to incorporate year-fixed effects and interactive terms of year- and country-fixed effects.

By doing so, the fixed-effect approach allows for the control of unobservable country-specific factors that remain constant over time. This approach effectively captures the individual heterogeneity among countries, providing a more robust analysis. On the other hand, to handle time-varying unobservable country characteristics, the inclusion of year-fixed effects and interaction

	terms allows for the investigation of how these attributes change over time and interact with country-specific effects. By incorporating these advanced statistical techniques, researchers can attain a more comprehensive understanding of the relationships and dynamics between the intervention (PEPFAR) and the observed outcomes, while appropriately accounting for various confounding factors that may influence the results. (3) However, it is crucial to clarify whether the timing of the PEPFAR intervention is indeed uniform across countries or if it varies. If the intervention's timing differs by country, the straightforward DID framework may not be appropriate. Instead, the authors should consider implementing event study and/or staggered DID estimates. Nonetheless, when utilizing staggered DID, it becomes imperative to confirm the satisfaction of three critical assumptions: parallel trend, no anticipation, and no spillover effects. Assessing and validating these assumptions is paramount, as they directly impact the validity and reliability of the study's findings. Only by ensuring that these assumptions are met can researchers draw accurate conclusions and confidently identify the true impact of the PEPFAR intervention on the observed outcomes across different countries and time periods. References for staggered DID Cengiz et al., QJE (2019) Sun-Abraham estimator (JE, 2020) Callway-Sant'Anna estimator (JE, 2020) de Chaisemartin-D'Haultfoeuille estimator (AER, 2020) Minor comments (1) The authors chose seven dependent variables from the World Bank's World Development Indicators as they serve as representative indicators of global health target areas. In the appendix, the rationale for this selection can be explained more extensively. These specific variables were likely considered because they provide essential insights into various aspects of global health and can effectively capture the impact of the PEPFAR intervention on key health-related outcomes. By focusing on these specific indicators, the study aims to offer a comprehensive assessment of the intervention's effects on multiple dimensions of global health, thereby contributing valuable information to the existing body of knowledge in this field. (2) Similarly, the rationale for the selected covariates can be elucidated in the appendix. Covariates are additional factors included in the analysis to control for potential confounding variables that might influence the relationship between the independent variable (PEPFAR intervention) and the dependent variables (global health outcomes). The authors likely selected covariates based on prior research, theoretical reasoning, or empirical evidence suggesting their relevance and influence on the outcomes of interest. By incorporating these covariates, the study aims to isolate the specific effect of the PEPFAR intervention on global health outcomes while accounting for other influential factors that might be at play.
--	---

	(3) The authors are encouraged to present the adjusted R-squared value, rather than the R-squared value, in the results section. The adjusted R-squared is a more appropriate measure of goodness-of-fit in multiple regression models. It takes into account the number of independent variables included in the model and provides a more accurate indication of how well the model explains the variance in the dependent variable while penalizing for overfitting. Reporting the adjusted R-squared value allows readers to better assess the model's fit and its ability to explain the observed variations in the dependent variables after controlling for the selected covariates.
--	---

REVIEWER	Pino, Eliana The University of Manchester, School of Health Sciences
REVIEW RETURNED	01-Aug-2023

GENERAL COMMENTS	Dear Authors, It was a pleasure reviewing your paper. I found the paper very interesting and an attempt to use impact evaluation methods in this type of literature. It provides a comprehensive analysis including many PEPFAR beneficiary countries. The authors also include various specifications in the supplementary material. My main suggestions are:  1. From the article it is not clear what is the need and motivation for writing the paper. What is the gap and how does the paper contribute to that? Is the use of quantitative methods novel? Is it the outcome variables? Is the data up to date? Is it the sub-group analysis? 2. It is important to explicitly acknowledge in the conflicts of interest that there was no involvement or collaboration with the Global Fund. Since the funding for the study is from the Global Fund and the cooperation and joint implementation of projects in PEPFAR and the Global Fund is globally recognised, I feel that as it stands it might leave the reader in doubt on the unbiasedness of the analyses. 3. I suggest using fixed effects to account for differences in time invariant country characteristics and report robust standard errors clustered at the level of the country group. I include other specific comments in the attachment. Hope to see you improving your manuscript and publishing it soon! (The reviewer provided a marked copy with additional comments. Please contact the publisher for full details.)
--

VERSION 1 – AUTHOR RESPONSE

Reviewer: 1

Dr. Aaron Richterman, Hospital of the University of Pennsylvania

Comments to the Author:

Authors report a difference-in-difference analysis assessing changes in country-level maternal and child health outcomes associated with a country being a PEPFAR recipient, with primary findings including improvements in vaccine-related outcomes, maternal mortality, and infant mortality. The conceptual justification for measuring spillover effects associated with HIV-related health aid is strong, but I found there to be a number of major issues, primarily methodological but also related to the classification of the study as an economic study and the research gap this study is supposed to be addressing, all outlined in detail below:

Title

R1.1. Why is this study called an economic evaluation, and why is the CHEERS checklist used if no costs are assessed? This seems to be an evaluation of health outcomes associated with a health aid program.

A1.1. We apologize for the classification error. The referee is correct that no cost or programmatic spending information is included in this paper. We have determined that none of the scientific guidelines apply to our paper.

Introduction

R1.2. Several key references are not included in the introduction or discussion — Bendavid Ann Intern Med 2010 (2009) (showing PEPFAR's effects on HIV-related outcomes), and particularly Bendavid JAMA 2012, showing reductions in all-cause mortality with PEPFAR.

A1.2. Thank you for pointing this out. We are familiar with the references, and we have cited them in other related papers. The Bendavid references in 2009, 2012, and 2016 have now all been included (see the last sentence of the 2nd paragraph in the Introduction section) and are discussed in the manuscript.

R1.3. Apart from those missed references, the authors summarize well the literature assessing the relationship between PEPFAR and maternal / child outcomes. I am left wondering what research gap motivates the current study — is it the size of the study in terms of countries included, or the timing, or something else? I recommend spelling out more clearly what the justification for this study is, particularly given the number of other analyses of country-level outcomes associated with PEPFAR.

A1.3. We have strengthened the paper by following up on your question. Indeed, there are four motivating factors for this research, now described better in the Introduction section.

1. The PEPFAR programming contains a number of explicit activities aimed at caring for pregnant women's health issues, and unborn and children's health. These programmatic activities of PEPFAR may well have impacts on health outcomes beyond HIV—and a literature of studies of such spillovers with PEPFAR has developed, apparently because of these programmatic features.
2. That literature on PEPFAR MCH spillovers shows conflicting findings on child mortality and on child immunizations. Maternal mortality has not been studied, though maternal care has been shown to be affected by PEPFAR.
3. Bendavid's team (2012) showed that total lives saved by PEPFAR exceeded the lives saved from HIV disease alone — suggesting spillover health benefits — though the difference was statistically insignificant.
4. Extensive PEPFAR funding of clinical staff, aimed at providing HIV services, has been shown to have substantial excess capacity (30-40%) (citation 24, Barnhart et al., 2019 and citation 25, Salihu, 2015). And the clinical sites have long been encouraged (by OGAC and country partners) to devote this excess capacity to treating other population health issues – which would create spillover impacts.

This situation warrants a study of PEPFAR spillover impacts throughout the 2004-2018 period. In addition, the study would extend the literature by estimating spillover impacts for separate cohorts of recipient countries (based on intensity of annual program planning activity and intensity of PEPFAR spending) to see if spillover impacts are conditional on intensity levels of funding and planning.

Our choice of measures of health are driven by prior use in HIV spillover studies.

Methods

R1.4. Related to #3, given prior work evaluating country-level associations between PEPFAR and maternal/child outcomes, this study would be greatly strengthened by considering individual-level outcomes measured by nationally representative surveys such as DHS (data are publicly available upon request). With DHS data, pregnancy-related and infant mortality can be estimated directly (see Bendavid JAMA 2012, Jakubowski PLoS Med 2017, Richterman Nature 2023), and infant vaccination history is also directly assessed.

A1.4. We are aware of DHS and other country health survey data (eg PHIA), and our team has done research using these data. While we agree that there are extraordinary opportunities to measure individual behaviors and outcomes from such data sets, our interest here was to examine a large number of recipient countries so that we could look into variations in PEPFAR policy across country cohorts (COP status, or not) (High, Medium and Low funding intensity levels). For that reason, we chose to use the World Bank measures of outcomes and the panel data set of country years. We agree that more research is going to be needed on spillover variations among population cohorts,

particularly related to equity (gender, low wealth, populations in rural areas, and other subpopulations)— and DHS/PHIA will need to be used for that kind of research.

R1.5. Some countries were selected as Pefpar recipient countries later than 2004, which makes a staggered design preferable, rather than pre-post 2004.

A1.5. We appreciate this comment very much, and are presenting staggered DID estimates in the revised manuscript side by side with traditional DID -- the consistency of results is readily apparent, which strengthens the paper. Presenting both is also useful since the essential pre-period assumptions are consistently met for DID, but for staggered DID there are important exceptions (1) the time trend assumptions required by staggered DID are not met for Maternal Mortality, and (2) for 3 immunizations (DPT, Measles and Tetanus) the assumptions are only met for the COP and high spend cohorts. See the Statistical Analysis section under Methods, and details on the specification testing for both the traditional and staggered DID models is provided in supplemental tables S3-S4.

R1.6. There is insufficient detail about the model specification. Are linear models used? Are cluster robust standard errors used? Are country and year fixed effects included? In the supplement, the variables in the equations are not defined.

A1.6. Thank you for this comment. In the revised manuscript, we applied both traditional and staggered DID approaches to estimate PEPFAR impacts. We have included detailed model specifications for each approach in the Methods section. For all models, we ran regressions with both unlogged and logged versions, presenting the results from the unlogged estimates as our main findings and the logged results in the supplemental sections S6, S8, and S10. The logged estimates were also used for robustness analysis. Additionally, robust standard errors were calculated for all regression analyses. In the staggered DID approach, we controlled for fixed effects at the year and country cohort levels.

Specifically, the staggered DID approach generalizes the traditional DID method by considering country-specific participation year and allows for dynamic leads and lags to the participation to be estimated. Equation (3) was used for staggered DID estimates:

$$MCH_{it} = \beta_0 + \beta_1 PEPFAR_i + \beta_2 D_{it} + \beta_3 X_{i,2004} + \gamma_t + \varepsilon_{it} \quad (3)$$

where D_{it} is a dichotomous variable that equals one for country i in each of its participation year and zero otherwise; β_2 is of our primary interest, measuring the impact of the presence of PEPFAR's support on recipient country MCH indicators; γ_t is a vector of year fixed effects dichotomous variables.

Specification testing of the staggered DID models was accomplished by estimating the dynamic impact of PEPFAR on each MCH outcome (see Supplement S4 for the results). This investigation serves two purposes: first, to test the essential assumption of the staggered DID method that determine whether the PEPFAR and comparison countries had similar pre-intervention outcome trends before recipient countries joined the program; and second, to track the evolution of PEPFAR's impact in terms of magnitude and statistical significance over time. We achieve this by substituting D_{it} in equation (3) with a set of country-specific dichotomous variables, D_{it}^T , indicating years relative to each country's PEPFAR initial participation year. Equation (4) was used:

$$MCH_{it} = \beta_0 + \beta_1 PEPFAR_i + \sum_{T=-10}^{15} \beta_T D_{it}^T + \delta X_{i,2004} + \gamma_t + \varepsilon_{it} \quad (4)$$

where D_{it}^T equals zero, except as following: D_{it}^T ($-10 \leq T \leq -1$) equals one for country i in its T th year before participating the PEPFAR program, while D_{it}^T ($0 \leq T \leq 15$) equals one for country i in its T th year after participating in the PEPFAR program. A set of statistically insignificant D_{it}^T ($-10 \leq T \leq -1$) without a discernible trend would confirm a similar pre-intervention trend between PEPFAR and comparison countries, demonstrating the internal validity of the staggered DID design. Further, we used the year prior to initial participation as the baseline year and therefore excluded the dummy variable of D_{it}^{-1} from the equation. We followed previous studies in using a 25-year window, which set D_{it}^{-10} equals one for all years that are 10 or more years before PEPFAR participation, while D_{it}^{15} equals one for all years that are 15 or more years after PEPFAR participation. Thus, PEPFAR's dynamic impacts estimated for these two endpoints may have greater variance and, consequently, less precision.

These details are now included in the Methods section of the manuscript. Full model results are reported in the supplemental materials S7-S10.

R1.7. There is no discussion or assessment of the parallel trends assumption, which is key to the difference-in-differences design.

A1.7. We appreciate the importance of this issue and thank you for raising it. We have tested all models (dependent variables by country cohorts) for baseline parallel trends between the treatment and control countries for the traditional DID models. Specification of these tests is described in our response to (6) above. The results of the tests show that the assumption is upheld for all PEPFAR cohorts. The pre-period assumptions for staggered DID are largely met, but there are important exceptions (1) the assumptions are not met for Maternal Mortality, and (2) for 3 immunizations (DPT, Measles and Tetanus) the assumptions are only met for the COP and High Spend cohorts. The supplemental sections S3 and S4 contain the results of all statistical tests of parallel baselines.

R1.8. Related to #7, temporal analyses (event study estimates) should be included.

A1.8. We agree. As noted in responses A1.6 and A1.7, we have added a parallel set of analyses using staggered DID with robust standard errors, along with extensive specification testing of both the traditional and staggered DID models.

We tested the staggered DID pre-trend assumption and the dynamic impact of the PEPFAR program by running an equation 4, which is intentionally specified within a temporal analyses (event study) framework. All the detailed information can be found in the revised Method section.

R1.9. There have been major advances in the difference-in-differences literature over the last five years, summarized well by Roth et al "What's Trending in Difference-in-Differences? A synthesis of the Recent Econometrics Literature" <https://doi.org/10.48550/arXiv2201.01194>. In particular there is risk of bias when intervention timing varies (as it does here) and the effect also changes over time. I would recommend including discussion of these issues and consider use of alternate approaches that are not vulnerable to this bias, again well-summarized in the Roth article.

A1.9. We appreciate your comment and have strengthened the manuscript in several ways. We now present Staggered DID as well as Traditional DID and have tested baseline trend assumptions for both. Please see responses A1.6 through A1.8.

R1.10. Why only consider baseline covariates, rather than time-varying?

A1.10. Our emphasis on baseline covariates rather than time varying covariates was a deliberate decision that was made to avoid endogeneity problems. Causal inference regarding PEPFAR's impact on spillovers is derived from a model specification designed to implement the Rubin potential outcomes framework (See citation XXX Imbens, 2021). We estimate the causal parameter, the average treatment effect (ATE) of PEPFAR, using both traditional and staggered differences-in-differences (DID) models, conditional on baseline non-financial covariates and baseline covariates on other (non-PEPFAR) donor spending on health and aggregate domestic spending on health (government and private) (see equation 1 and equation 3).

R1.11. How was high, medium, and low PEPFAR disbursements defined?

A1.11. High, medium and low country cohorts were selected as three tertiles of 30 countries each (highest cumulative PEPFAR spending per capita 2004-2018, medium 30 countries, lowest 30 countries). See the Study Design section in Methods.

Reviewer: 2

Dr. Haruko Noguchi, Waseda University

Comments to the Author:

Overall evaluation

This study endeavors to assess the spillover effects of the US President's Emergency Plan for AIDS Relief (PEPFAR) on maternal and child health outcomes. To achieve this, the authors employ the difference-in-differences method (DID) on a longitudinal dataset spanning 29 years (1990-2018) comprising 157 low- and middle-income countries (LMICs). Within this dataset, they distinguish 90 PEPFAR recipient countries from 67 countries that either received no PEPFAR support or minimal PEPFAR support. This study provides compelling evidence that the presence of PEPFAR has consistently and positively influenced critical health indicators beyond HIV, including child and maternal mortality rates, as well as childhood immunization rates.

Given the inconclusive nature of previous studies regarding the spillover effects of PEPFAR, this study aims to address this knowledge gap by undertaking a comprehensive program evaluation that encompasses seven key health outcome measures. These measures have been thoughtfully selected from the World Bank's World Development Indicators, which serve as a representation of global health target areas. The inclusion of these diverse health indicators enhances the scope and rigor of the analysis, allowing for a more comprehensive assessment of PEPFAR's impact on various health aspects beyond its primary focus on HIV/AIDS.

While I acknowledge the importance of this study, I believe that the current manuscript lacks a comprehensive elucidation of the institutional context of PEPFAR and the underlying hypothetical mechanisms driving the observed spillover effects. Additionally, I would like to highlight certain econometric methodological issues that need to be addressed in this study.

I kindly request you to refer to the detailed comments provided below.

2. Major Comments

R2.1. The elucidation of the institutional context and the consequent hypothetical mechanism remains unclear:

Initially, the authors presented a concise overview of PEPFAR and reviewed relevant literature focusing on its spillover effects, summarizing these details in the "Introduction" section. However, for readers who are not well-acquainted with PEPFAR, a comprehensive understanding of the potential mechanisms linking PEPFAR's interventions with other health promotion programs, and thereby leading to unintended and indirect effects (whether positive or negative) on various health outcomes, remains elusive.

For instance, the authors characterized PEPFAR as a "vertical" intervention. This term refers to an approach that addresses a specific health issue in isolation, often through targeted programs. How this would cause "crowd out" of scarce resources in detail? How does PEPFAR contribute to situations where women not infected with HIV end up giving birth in areas with more HIV-funded patient service or greater HIV infrastructure support? Furthermore, there is ambiguity about how PEPFAR facilitates the advancement of vaccination programs or initiatives like the Prevention of Mother-to-Child Transmission (PMTCT) program and the Saving Mothers, Giving Life program.

In light of these concerns, I propose that the authors furnish a more detailed account of PEPFAR, elucidating the hypothetical mechanisms that underpin the observed spillover effects. If the word limit is insufficient, kindly consider relocating all pertinent information of PEPFAR to the appendix section for comprehensive coverage.

A2.1. Thank you for pointing out the need for a background on how the program works, and what motivates the study of health spillovers. We have included in the Introduction section a very general description of PEPFAR programming and some specific mechanisms through which the program may be creating spillover health effects.

Reviewer #1 raised a similar concern about the rationale for the study. Your focus is on the mechanisms of the intervention that might generate spillovers to maternal and child health. We expand here to our response to reviewer #1.

The large and long-lasting health care aid program contains a number of possible mechanisms for producing health spillovers beyond preventing and treating HIV disease. Five possible mechanisms are discussed in the text.

1. The first is significant PEPFAR spending on staffing and other operating expenses required to operate health clinics and testing services in every country. But, this significant clinic capacity for delivering HIV services has clearly exceeded the demand in most locations. Emerging evidence from activity-based costing and management studies across multiple countries indicates that PEPFAR funded clinicians who are fully funded to do HIV work actually spend 30-40% of their time providing broader primary health care services at the facility level (see citation 24, Barnhart et al., 2019 and citation 25, Salihi, 2015). PEPFAR's State Department Oversight Agency (OGAC) and country partners have been encouraging the facility managers, staff, and country program officials to expand the scope of offered services in these facilities beyond those services related to HIV in order to provide a valuable public service to otherwise underserved populations. It seems likely that many or most of these non-HIV services would not have been provided to households in the absence of PEPFAR.

2. The second spillover mechanism is prevention of mother to child transmission of HIV activities (PMTCT), extensively supported by PEPFAR and other donor programs in LMICs. These programs seek out pregnant women, provide professional health services in support of better maternal and child outcomes, including HIV testing and treatment for pregnant women and various prevention services

for unborn children. These programs are very widespread, and along with other health professional services aimed at women and children, may have introduced women to professional (and free) health services for the first time. This may well have sparked changed household behavior to subsequently use more of these professional health services for themselves and their children for non-HIV related issues. Not only better birthing care outcomes may be the result, but also access to vaccinations and other prevention regimes for their children.

3. Other PEPFAR activities like building/operating more accessible facilities for testing, treatment, and sponsoring visible community programming for sensible risk behavior change may well have also led to attitude changes about trained medical professionals, and improved trust in them to deal with health issues for themselves and their children.

4. PEPFAR spending has certainly also boosted the economy and jobs in these poor countries. The level of program spending is a significant influence in these poor countries. Other research (citation 34 Crown et al., 2023 and citation 35 Wagner et al., 2015) shows that the effects of PEPFAR activities has boosted incomes and jobs. Considerable PEPFAR programming is also aimed at training over 340,000 health workers to work in the facilities.

5. Finally, the program introduced an objective in 2008 of strengthening domestic health systems in recipient countries. This mechanism may have improved government health programming for many kinds of health issues, and may have also improved access for rural and poor families to health services. One prominent research team commented on the health system strengthening initiative:” In particular, PEPFAR became involved in building new medical education systems, a substantial departure from its core areas of experience. Efforts to demonstrate that PEPFAR’s core activities – supporting ART and HIV care and prevention programs – yielded large positive spillover benefits beyond HIV (citation 4, Bendavid 2016).

We have added a section, Background on Possible Spillover Mechanisms in the Introduction section, to the paper discussing these plausible mechanisms of spillovers. We have also noted, in the Discussion section, that research is needed to identify the importance of these items (and other mechanisms like allocations of funds) in creating non-HIV health effects.

R2.2. Econometric Strategy: necessary assumptions to be met for DID and timings of intervention:

(1) Initially, when employing a simple DID analysis, adherence to the parallel trend assumption becomes indispensable. Upon meticulous examination of Figures 1 and 2, the presence of this crucial assumption appears inconclusive. To ascertain its validity, the authors should conduct a rigorous statistical assessment, comparing outcome disparities before the intervention's temporal demarcation in 2004. Specifically, this evaluation should encompass the COP PEPFAR versus non-PEPFAR groups, as well as the ALL PEPFAR versus non-PEPFAR groups.

To accomplish this verification, the authors can undertake the reconstruction of Figures 1 and 2, delineating the distinctions observed in raw data alongside their respective 95% confidence intervals. Crucially, scrutinizing whether these confidence intervals envelop the null point (0) and encompass both positive and negative values prior to 2004 will elucidate the adherence to the parallel trend assumption. Should the confidence intervals encompass 0 and span both directions, it indicates the fulfillment of the parallel trend assumption, denoting the absence of statistically significant outcome disparities between the aforementioned groups preceding the intervention. Conversely, the presence of confidence intervals diverging from 0, while straddling either the positive or negative domains,

would signify statistical significance in outcome differences prior to the intervention. Such circumstances would suggest a potential violation of the parallel trend assumption, thereby rendering the results susceptible to bias in the DID analysis.

A2.2-1. Thank you for offering guidance on this critical test for traditional DID. This analysis of parallel trends has been done (Results are in Supplemental Sections S3 and S4). We find that parallel trends do exist for all country cohorts modeled for all outcomes. The tests for pre-period trends were also done for the staggered DID. As described elsewhere, the staggered DID pre-period assessment concludes that while generally met, there are important situations where staggered DID does not meet the critical pre-period assumption. Specifically, no country cohorts meet the pre-trend assumption for maternal mortality rate. And for 3 vaccination outcomes (DPT, Measles, and Tetanus) the pre-period assumptions are met for only the COP and the high-spend cohorts. The Supplemental materials provide the results of these and all other tests.

R2.2-2 Secondly, I hold reservations regarding the assumption that the intervention of PEPFAR occurs at random. Despite the authors' efforts to control for various country characteristics during their analysis, I strongly recommend confirming whether the differences in these characteristics between the treatment (PEPFAR-recipient) and control (non-PEPFAR-recipient) groups are statistically insignificant. In essence, it is essential to ascertain if the attributes of both groups are well-balanced. In the absence of balance, adopting a DID combined with a propensity score matching method (PPS) would be prudent. This ensures a more rigorous evaluation of the intervention's impact and mitigates potential biases that might arise from unbalanced characteristics between the two groups.

Therefore, I advise the authors to thoroughly examine and statistically validate the comparability of country attributes between the PEPFAR and non-PEPFAR groups. If imbalances are detected, the implementation of PPS would be a prudent course of action to attain a more robust and credible analysis, thereby enhancing the reliability of the study's outcomes, although PPS could adjust only observable factors.

In addition to the PSM framework, to account for time-invariant unobservable country characteristics, one may consider employing a fixed-effect approach by leveraging the panel data. Furthermore, to address time-variant unobservable country attributes, it is essential to incorporate year-fixed effects and interactive terms of year- and country-fixed effects.

By doing so, the fixed-effect approach allows for the control of unobservable country-specific factors that remain constant over time. This approach effectively captures the individual heterogeneity among countries, providing a more robust analysis. On the other hand, to handle time-varying unobservable country characteristics, the inclusion of year-fixed effects and interaction terms allows for the investigation of how these attributes change over time and interact with country-specific effects.

By incorporating these advanced statistical techniques, researchers can attain a more comprehensive understanding of the relationships and dynamics between the intervention (PEPFAR) and the observed outcomes, while appropriately accounting for various confounding factors that may influence the results.

A2.2-2. Thanks for these suggestions. We followed these suggestions and conducted propensity score matches with the expectation that the observable characteristics between the recipient and comparison countries could be better balanced before we ran DID analyses. We used the full set of baseline covariates for the match and tried multiple matching strategies, including the nearest neighbor match with different distances and a kernel match approach. However, all the match strategies showed poor common support results. Take hepatitis B immunization rate as an example. For the COP country cohort, more than one-third of the treated countries could not be successfully matched with control countries in the nearest neighbor match approach. Due to the poor performance of the PSM common support results, we felt that the traditional and staggered DID approaches with robust standard errors conditional on baseline covariates provided the strongest analytic design. Rigorous specification testing of the assumptions of these models provide insights into the potential for bias across the models for the various subcohorts of countries.

We followed your suggestion of introducing a set of dichotomous variables in the staggered DID specifications to control for fixed-effects at the year and country cohort levels. Details on this model specification are described in our response A1.6 to reviewer 1 as well as the Covariates section in Methods.

R2.2-3. However, it is crucial to clarify whether the timing of the PEPFAR intervention is indeed uniform across countries or if it varies. If the intervention's timing differs by country, the straightforward DID framework may not be appropriate. Instead, the authors should consider implementing event study and/or staggered DID estimates. Nonetheless, when utilizing staggered DID, it becomes imperative to confirm the satisfaction of three critical assumptions: parallel trend, no anticipation, and no spillover effects.

A2.2-3. We have implemented a staggered DID approach, in log and unlogged forms. Tables 2 and 3 in the text show the unlogged staggered model results (the full model results for the unlogged and logged staggered DID are presented in supplemental materials S9 and S10). The staggered DID models show the same overall pattern as the original DID models (significant and consistent spillover impacts for child and maternal mortality, and much smaller impacts on immunizations, and effects on women's anemia only in highly funded or COP countries).

We tested the parallel trend assumption for the staggered DID approach by regressing equation (4), which is intentionally specified in an event study framework. We did not test for anticipation and spillover effects because we assume that the performance of maternal and child health outcomes in a later treated country is unlikely to be influenced by other LMICs who received PEPFAR's support at an early stage.

Assessing and validating these assumptions is paramount, as they directly impact the validity and reliability of the study's findings. Only by ensuring that these assumptions are met can researchers draw accurate conclusions and confidently identify the true impact of the PEPFAR intervention on the observed outcomes across different countries and time periods.

References for staggered DID

Cengiz et al., QJE (2019)

Sun-Abraham estimator (JE, 2020)

Callway-Sant'Anna estimator (JE, 2020)

de Chaisemartin-D'Haultfoeuille estimator (AER, 2020)

We appreciate this list of references. We cited two significant papers for the staggered DID method:

38. Clarke, D., & Tapia-Schyte, K. (2021). Implementing the panel event study. *The Stata Journal*, 21(4), 853-884.

39. Beck, T., Levine, R., & Levkov, A. (2010). Big bad banks? The winners and losers from bank deregulation in the United States. *The journal of finance*, 65(5), 1637-1667.

Minor comments

R2.3-1. The authors chose seven dependent variables from the World Bank's World Development Indicators as they serve as representative indicators of global health target areas. In the appendix, the rationale for this selection can be explained more extensively. These specific variables were likely considered because they provide essential insights into various aspects of global health and can effectively capture the impact of the PEPFAR intervention on key health-related outcomes. By focusing on these specific indicators, the study aims to offer a comprehensive assessment of the intervention's effects on multiple dimensions of global health, thereby contributing valuable information to the existing body of knowledge in this field.

A2.3-1. The text now includes our rationale for these indicators. The child mortality and immunizations have been used in the spillover literature where results have been inconclusive about PEPFAR impact. Maternal mortality and anemia are widely used in the studies of maternal and obstetric care literature. Please see citations 12,13,14,15,16,17,18,19.

R2.3-2. Similarly, the rationale for the selected covariates can be elucidated in the appendix. Covariates are additional factors included in the analysis to control for potential confounding variables that might influence the relationship between the independent variable (PEPFAR intervention) and the dependent variables (global health outcomes). The authors likely selected covariates based on prior research, theoretical reasoning, or empirical evidence suggesting their relevance and influence on the outcomes of interest. By incorporating these covariates, the study aims to isolate the specific effect of the PEPFAR intervention on global health outcomes while accounting for other influential factors that might be at play.

A2.3-2. We provide a table in the supplementary S2, identifying all baseline (2004) covariates, and for each, a brief reason for including it to help balance the control and treatment groups.

R2.3-3 The authors are encouraged to present the adjusted R-squared value, rather than the R-squared value, in the results section. The adjusted R-squared is a more appropriate measure of goodness-of-fit in multiple regression models. It takes into account the number of independent variables included in the model and provides a more accurate indication of how well the model explains the variance in the dependent variable while penalizing for overfitting. Reporting the adjusted R-squared value allows readers to better assess the model's fit and its ability to explain the observed variations in the dependent variables after controlling for the selected covariates.

A2.3-3. Due to space limitations in the manuscript table reporting the estimated treatment effects across model cohorts and specifications we were not able to include R2 or adjusted R2 values in the main manuscript results. However, full model results, including adjusted R2 are reported in the supplemental materials S7-S10.

Reviewer: 3

Dr. Eliana Pino, The University of Manchester

Comments to the Author:

Dear Authors,

It was a pleasure reviewing your paper. I found the paper very interesting and an attempt to use impact evaluation methods in this type of literature. It provides a comprehensive analysis including many PEPFAR beneficiary countries. The authors also include various specifications in the supplementary material.

My main suggestions are:

R3.1. From the article it is not clear what is the need and motivation for writing the paper. What is the gap and how does the paper contribute to that? Is the use of quantitative methods novel? Is it the outcome variables? Is the data up to date? Is it the sub-group analysis?

A3.1. This comment is very helpful—and the text now includes additional explanation of how PEPFAR works, the rationale for how PEPFAR might generate health spillover effects, and a paragraph on the conflicting findings of previous studies of PEPFAR spillovers -- all making the case for this study..

R3.2. It is important to explicitly acknowledge in the conflicts of interest that there was no involvement or collaboration with the Global Fund. Since the funding for the study is from the Global Fund and the cooperation and joint implementation of projects in PEPFAR and the Global Fund is globally

recognised, I feel that as it stands it might leave the reader in doubt on the unbiasedness of the analyses.

A3.2. We have written an acknowledgement of funding for our work, recognizing OGAC, Gates and the GF— and explicitly saying the study was designed and implemented at Brandeis.

R3.3. I suggest using fixed effects to account for differences in time invariant country characteristics and report robust standard errors clustered at the level of the country group.

I include other specific comments in the attachment.

Hope to see you improving your manuscript and publishing it soon!

A3.3. Thank you for your comments. We followed your suggestions and calculated robust standard errors for all the estimates. We have included a set of dichotomous variables to control for the fixed effects at the year and country cohort levels in the staggered DID setting (Please see our responses A1.5, A.1.6, A1.8-A.1.10 and A2.2-1 and A2.2-2 regarding fixed effects and our addition of staggered DID analyses to the manuscript).